Identification of a circRNA-miRNA-mRNA regulatory network for exploring novel therapeutic options for glioma

He Yi 1
Chen Yihong 1
Tong Yuxin 2 3
Long Wenyong wylongdr@csu.edu.cn 1
Liu Qing liuqingdr@csu.edu.cn 1
1 Neurosurgery Department, Xiangya Hospital Central South University , Changsha , Hunan , China
2 Department of Ophthalmology, Second Xiangya Hospital, Central South University , Changsha , Hunan , China
3 Hunan Clinical Research Center of Ophthalmic Disease , Changsha , Hunan , China
Shang Yuan
Electronic publication date: 2021 Aug 6
Publication date: 2021
Volume: 9
Electronic Location ID: e11894
Received 2021 May 21; Accepted 2021 Jul 12
Copyright: ©2021 He et al.
Copyright year: 2021
Copyright holder: He et al.
License: This is an open access article distributed under the terms of the Creative Commons Attribution License, which permits unrestricted use, distribution, reproduction and adaptation in any medium and for any purpose provided that it is properly attributed. For attribution, the original author(s), title, publication source (PeerJ) and either DOI or URL of the article must be cited.
License URL: https://creativecommons.org/licenses/by/4.0/

Keywords: Glioma, circRNA, ceRNA network, Novel therapy, Biomarker

Funding: National Natural Science Foundation of China 81802974 This work was supported by the National Natural Science Foundation of China (grant number 81802974). The funders had no role in study design, data collection and analysis, decision to publish, or preparation of the manuscript.

==============================
Background

Glioma is the most common brain neoplasm with a poor prognosis. Circular RNA (circRNA) and their associated competing endogenous RNA (ceRNA) network play critical roles in the pathogenesis of glioma. However, the alteration of the circRNA-miRNA-mRNA regulatory network and its correlation with glioma therapy haven’t been systematically analyzed.

Methods

With GEO, GEPIA2, circBank, CSCD, CircInteractome, mirWalk 2.0, and mirDIP 4.1, we constructed a circRNA–miRNA–mRNA network in glioma. LASSO regression and multivariate Cox regression analysis established a hub mRNA signature to assess the prognosis. GSVA was used to estimate the immune infiltration level. Potential anti-glioma drugs were forecasted using the cMap database and evaluated with GSEA using GEO data.

Results

A ceRNA network of seven circRNAs (hsa_circ_0030788/0034182/0000227/ 0018086/0000229/0036592/0002765), 15 miRNAs(hsa-miR-1200/1205/1248/ 1303/3925-5p/5693/581/586/599/607/640/647/6867-5p/767-3p/935), and 46 mRNAs (including 11 hub genes of ARHGAP11A, DRP2, HNRNPA3, IGFBP5, IP6K2, KLF10, KPNA4, NRP2, PAIP1, RCN1, and SEMA5A) was constructed. Functional enrichment showed they influenced majority of the hallmarks of tumors. Eleven hub genes were proven to be decent prognostic signatures for glioma in both TCGA and CGGA datasets. Forty-six LASSO regression significant genes were closely related to immune infiltration. Finally, five compounds (fulvestrant, tanespimycin, mifepristone, tretinoin, and harman) were predicted as potential treatments for glioma. Among them, mifepristone and tretinoin were proven to inhibit the cell cycle and DNA repair in glioma.

Conclusion

This study highlights the potential pathogenesis of the circRNA-miRNA-mRNA regulatory network and identifies novel therapeutic options for glioma.

Introduction

Gliomas comprise majority of primary intracranial neoplasms with high heterogeneity and aggressiveness, resulting in a poor prognosis even after current standard combination treatments (Ostrom et al., 2018b; Wesseling & Capper, 2018). The five-year survival rate is only approximately 5% (Ostrom et al., 2018a; Ostrom et al., 2018b; Wesseling & Capper, 2018). Recent advances in precision medicine, genomics, immunology, and other disciplines have uncovered multiple experimental therapies, such as targeted therapy, gene therapy, immunotherapy, and novel drug-delivery technologies that could possibly shed light on the treatment strategies for gliomas (Lapointe, Perry & Butowski, 2018). Therefore, it is of great importance to explore the internal mechanisms of gliomas to identify new therapeutic targets.

Circular RNA (circRNA) is a type of non-coding RNA derived from the exon or intron region of a gene (Kristensen et al., 2019). Since there is no 5–3 polarity and polyA tail, circRNAs are more stable than linear RNAs (Kristensen et al., 2019; Meng et al., 2017). CircRNA regulates the expression of a series of genes by modulating every stage of mRNA metabolism, including sequestration of microRNAs (miRNA) or proteins, modulation of transcription, interference with splicing, and translation to produce polypeptides (Chen, 2020; Wu et al., 2020). During the sequestration of miRNAs, circRNAs act as molecular sponges for miRNAs through their miRNA response elements (MREs), thereby de-repressing all target genes of the respective miRNA family (Hansen et al., 2013). Recently, circRNAs have been found to participate in multiple tumor phenotypes, including proliferation, invasion, metabolism, and immune response, making them promising diagnostic and prognostic markers as well as therapeutic targets for cancers (Chen, 2020).

Recent studies have shown that instead of pure malignant cells, the core tumors are actually surrounded by a complex microenvironment, including the immune cells (Binnewies et al., 2018). Immune cells infiltrating the tumor microenvironment have been confirmed with the ability to predict the patients’ clinical outcomes and also the efficacy of immunotherapy. Therefore, identifying immune cells infiltration, especially their pattern correlated with specific genes and gene signatures, is of great significance for estimation of the prognosis of GBM patients and the value of various therapies (Ali et al., 2016; Quail & Joyce, 2013).

Here, by utilizing bioinformatics methods, we identified differentially expressed circRNAs (DECs) in gliomas and studied their functions in gliomas as competing endogenous RNAs (ceRNAs). The workflow diagram is shown in Fig. 1, where DECs were first acquired from the circRNA-related microarray datasets of gliomas in the GEO database. Then, we forecasted and collected their related miRNAs and their corresponding target genes and built a circRNA-miRNA-mRNA regulatory network. Functional enrichment analyses were performed to determine their potential roles in the pathogenesis of gliomas. Furthermore, the hub genes were obtained through LASSO and multivariate Cox regression analyses and evaluated with ROC curve analysis and K-M curve analysis. Subsequently, GSVA analysis was performed to determine their correlation with immune infiltration. Finally, a connectivity map (CMap) was used to predict corresponding bioactive compounds and potential drugs for treatment, which were further assessed with GSEA in GEO database.

Figure 1 Workflow diagram of the construction of a circRNA-associated ceRNA network, identification and evaluation of a hub gene signature, and the prediction of potential therapeutic options for glioma.

Materials & Methods

Data obtained and DECs acquired

Microarray circRNA expression profile data of gliomas and corresponding normal tissues were screened and acquired from the Gene Expression Omnibus (GEO, https://www.ncbi.nlm.nih.gov/geo/) database, which is a public functional genomic database that allows users to query, locate, review, as well as download research and gene expression profiles (Barrett et al., 2013). DECs in the GSE109569 and GSE146463 datasets were analyzed and identified using GEO2R with the criteria of —log2 (fold change)—> 1 and P value < 0.05. The circRNAs upregulated or downregulated in both datasets were selected for further analysis.

Prediction of MREs

We employed three public databases to predict the MREs of the selected DECs: CircBank (http://www.circbank.cn/index.html) is a comprehensive database with more than 140,000 human-annotated circRNAs from different sources, providing abundant information on circRNAs, including their predicted binding miRNAs (Liu et al., 2019). The cancer-specific circRNA database (CSCD, http://gb.whu.edu.cn/CSCD/) is a cancer-specific circRNA database incorporating more than 272,000 cancer-specific circRNAs, with the aim of predicting MRE sites, RNA binding protein (RBP) sites, and open reading frames (ORFs) for each circRNA (Xia et al., 2018). Circular RNA Interactome (CircInteractome, http://circinteractome.nia.nih.gov) is a web tool for mapping RBP and MRE sites on human circRNAs by searching public databases of circRNA, miRNA, and RBP. It has multiple functions, including identifying potential circRNAs that can act as RBP sponges (Dudekula et al., 2016). An overlap in at least two databases was the basis for considering candidate target miRNAs of these DECs used for further mRNA prediction. The regulatory roles of these miRNAs and related regulation pathways were assessed using DIANA-miRPath v3.0 (http://snf-515788.vm.okeanos.grnet.gr/), which is a powerful online software for functional analysis of miRNAs (Vlachos et al., 2015).

Forecasting miRNA–mRNA interactions

miRNA–mRNA interactions were predicted using two integrated miRNA databases. MiRWalk 2.0 (http://zmf.umm.uni-heidelberg.de/mirwalk2) is a web tool that provides information about validated or putative miRNA–mRNA interactions. For its prediction, 12 algorithms (miRWalk, Microt4, mirbridge Targetscan, RNAhybrid, RNA22, PITA, Pictar2, miRNAMap, miRDB, miRanda, and miRMap) were employed to ensure robustness (Dweep & Gretz, 2015). Here, targeted genes forecasted by at least seven algorithms, along with the validated genes, were selected as candidate genes from miRWalk 2.0. Meanwhile, mirDIP v4.1 (http://ophid.utoronto.ca/mirDIP/) is a miRNA database integrated across 30 different resources, capable of providing nearly 152 million human microRNA–target predictions (Tokar et al., 2018). In the mirDIP v4.1 database, genes predicted by at least 11 algorithms under the very high score class were selected as candidate genes from mirDIP v4.1.

Obtaining DEGs and overlapped target genes

The Gene Expression Profiling Interactive Analysis (GEPIA) web server is a valuable resource for gene expression analysis based on tumor and normal samples from the TCGA and GTEx databases. GEPIA2 is an updated and enhanced version with higher resolution and more functionalities (Tang et al., 2019). Through GEPIA2, We identified differentially expressed genes (DEGs) between glioblastoma (GBM) and normal tissues using the criteria of —log2 (fold change)—> 1 and P value < 0.01. These DEGs were intersected with the candidate gene sets from miRWalk 2.0 and mirDIP v4.1. Overlapped mRNAs showing up in all three sets were taken as final target mRNAs and used for further analysis.

Functional enrichment analysis of overlapped genes

The Search Tool for the Retrieval of Interacting Genes database (STRING) is a database aimed at achieving a comprehensive and objective global network, including direct (physical) and indirect (functional) interactions (Szklarczyk et al., 2019). It was utilized to perform Gene Ontology (GO) analysis and Kyoto Encyclopedia of Genes and Genomes (KEGG) pathway enrichment analysis for the overlapped mRNA, with a setting P < 0.05 and counts > 5.

Identification and assessment of hub genes

Using mRNA expression profiles and clinical information from TCGA (https://www.cancer.gov/tcga), the overlapping genes were consecutively analyzed with LASSO regression and multivariate Cox regression analysis, and independent prognostic genes were identified as hub genes. The total risk score of each sample was calculated as the sum of the multiplication of the expression value and the correlation coefficient of each gene. Patients with higher 50% or lower 50% of risk score were defined as high-risk or low-risk groups respectively. Their value as a prognostic signature for gliomas, as well as their corresponding sensitivity and specificity, were evaluated using K-M curve analysis and ROC curve analysis both in the TCGA training dataset and the external CGGA validation dataset (http://www.cgga.org.cn/) (Zhao et al., 2021). Sample IDs of the CGGA samples used in this study was listed in Table S4. The protein level expression differences of hub genes were further confirmed with immunohistochemistry (IHC) images from The Human Protein Atlas (HPA) database (Uhlen et al., 2015).

Construction of a circRNA–miRNA–mRNA network

Cytoscape is an open-source software for the integration of molecular interaction network data and the establishment of powerful visualization (Shannon et al., 2003). Here, it was used to construct a circRNA–miRNA–mRNA regulatory network.

Assessment of immune cell infiltration

Gene set variation analysis (GSVA) is a gene set enrichment method and an open source software package for R, which can estimate the variation of pathway activity over a sample population in an unsupervised manner (Hanzelmann, Castelo & Guinney, 2013). To estimate the immune cell infiltration level, we applied single-sample gene-set enrichment analysis (ssGSEA), which is a built-in algorithm of the GSVA package, using the RNA-seq data and related clinical data from the TCGA-GBMLGG dataset. Gene expression features of 24 immune cells were acquired from a previous study (Bindea et al., 2013), and the correlation with immune cell infiltration was obtained for the genes that passed LASSO regression analysis. Sample IDs of the TCGA-GBMLGG samples used in this study was listed in Table S5.

Connectivity Map (CMap) analysis and assessment

The connectivity map (CMap) is a collection of genome-wide transcriptional expression data from human cell lines treated with various drugs or compounds. Functional connections between drugs, genes, and diseases were then uncovered using pattern-matching algorithms and features of common gene expression changes (Lamb, 2007; Lamb et al., 2006). Using the hub genes from multivariate Cox regression analysis, candidate compounds with negative connectivity scores were identified as promising candidate therapeutic approaches. The available related RNA-seq data were acquired from GEO and used for GSEA analysis to explore the effects of those drugs on gliomas.

Statistical analysis

R software (version 4.0.3) was used for all statistical analyses, and p-values <0.05 were considered statistically significant. The Glmnet package and survival package were utilized for LASSO regression and multivariate Cox regression analysis, respectively. GGally and rms packages were used to evaluate and remove the co-linearity between samples. K-M curve analysis was performed using the Survminer package. ROC curve analysis was performed using the survival ROC package. Visualization was achieved using the ggplot2, pheatmap, or plotROC packages.

Results

Acquiring eight DECS in gliomas

To explore the potential function of circRNAs and the corresponding ceRNA network in glioma, DECs from GSE109569 (three glioma samples vs. three normal samples) and GSE146463 (eight glioma samples vs. three normal samples) datasets were obtained using GEO2R from the GEO database. Genes with P < 0.05 and a —Log2(fold change)—> 1 were considered significant DECs. Through the intersection of two DEC datasets, six upregulated and two downregulated circRNAs were identified and chosen as research objects in this study. The differences in expression between gliomas and normal tissues are shown in Figs. 2A–2B. The basic features of these eight circRNAs are listed in Table 1. The circRNAs’ accurate expression values in the GEO datasets are summarized in Table S1.

Figure 2 Expression profile heatmaps for 8 DECs in two GEO datasets (A, B) and basic structures of the circRNAs (C).

The different colors and shapes in the outer and inner ring represent the different exons and the positions of MRE, RBP and ORF.

Identification of circRNA–miRNA interactions

To explore the roles of these eight circRNAs as ceRNAs in glioma, three online databases, namely circBank, CSCD, and CircInteractome, were utilized to collect potential target miRNAs. Six out of eight circRNAs were recorded in the CSCD database, and their structures of MRE, RBP, and ORF are shown in Fig. 2C. A total of 15 miRNAs and 18 circRNA–miRNA interactions were identified by at least two databases, including hsa_circ_0030788-miR-5693/miR-6867-5p/miR-607/miR-1248/miR-586/miR-599, hsa_circ_0034182-miR-1200, hsa_circ_0000227-miR-647/miR-1303/miR-767-3p, hsa_circ_0018086-miR-1303/miR-3925-5p/miR-581, hsa_circ_0000229-miR-935/miR-640, hsa_circ_0036592-miR-1205/miR-767-3p, and hsa_circ_0002765- miR-587/miR-767-3p. The functions of these miRNAs in tumors reported in PubMed were summarized in Table 2. Among them, miR-1303, miR-581, miR-586, miR-599, miR-607, miR-647, miR-767-3p, miR-935, and miR-1248 have been extensively reported to regulate the progression of various tumors as promoters or suppressors. DIANA-miRPath was then used to probe signaling pathways involving 15 unique miRNAs. As shown in Fig. 3A, these miRNAs are involved in multiple pathways of glioma, including the FoxO signaling pathway, phosphatidylinositol signaling pathway, and TGFβ signaling pathway. Furthermore, given that the expression levels of the miRNAs are critical for their biological functions, we checked the expression levels of those 15 miRNAs in glioma tissues with CGGA data and GEO data, as shown in (Tables S2–S3) Different data sets showed slightly different expression of those miRNAs. And their expression values vary quite a lot from sample to sample. But in general, according to those two datasets, the miR607 and miR587 have relatively lower expression levels.

Table 1 Basic features of Differentially expressed circRNAs.

circBase ID	bestTranscript	Position	strand	Length	Host gene Symbol	circRNA type	Regulation	
hsa_circ_0001156	NM_015568	chr20: 37547116-37547282	+	166	PPP1R16B	exonic	down	
hsa_circ_0030788	NM_052867	chr13: 101997616-102031004	−	508	NALCN	exonic	down	
hsa_circ_0034182	NM_000814	chr15: 26825465-26828561	−	221	GABRB3	exonic	down	
hsa_circ_0000227	NM_030751	chr10: 31644072-31676195	+	32123	ZEB1	intronic	up	
hsa_circ_0018086	NM_001128128	chr10: 31676052-31676195	+	143	ZEB1	intronic	up	
hsa_circ_0000229	NM_030751	chr10: 31661946-31709678	+	47732	ZEB1	intronic	up	
hsa_circ_0036592	NR_004859	chr15: 85180577-85181708	+	156	SCAND2	exonic	up	
hsa_circ_0002765	NM_001128128	chr10: 31644075-31676727	+	32652	ZEB1	intronic	up	

Table 2 Functions of miRNAs identified for ceRNA network.

predicted upstream hsa_circ_#	has-miR- #	Tumor type	Regulation axis	Role of miRNA in tumor	Ref.	
0000227	1303	Breast Cancer	miR-1303/CDKN1B	Promotor	Chen et al. (2020b)	
Breast Cancer	HIF-1 α/lncRNA-BCRT1/miR-1303/PTBP3	Suppressor	Liang et al. (2020)	
Gastric Cancer	miR-1303/CLDN18	Promotor	Zhang et al. (2014)	
Neuroblastoma	miR-1303/GSK3 β	Promotor	Li et al. (2016)	
0018086	581	Colorectal Cancer	miR-581/SMAD7	Promotor	Zhao et al. (2020)	
Hepatocellular Carcinoma	miR-581/EDEM1	Promotor	Wang et al. (2014)	
0030788	586	Cervical Cancer, Colon Cancer, etc.	lncRNA-MIF/miR-586	Promotor	Zhang et al. (2016)	
Osteosarcoma	miR-586	Promotor	Yang et al. (2015)	
0030788	599	Anaplastic Thyroid Carcinoma	lncRNA-NEAT1/miR-599	Suppressor	Tan et al. (2020)	
Esophageal Carcinoma	circ_0030018/miR-599/ENAH	Suppressor	Wang et al. (2019b)	
Esophageal Carcinoma	HIPK3/miR-599/c-MYC	Suppressor	Ba et al. (2020)	
Gastric Cancer	circ_0008035/miR-599/EIF4A1	Suppressor	Li et al. (2020)	
Gastric Cancer	miR-599/EIF5A2	Suppressor	Wang et al. (2018)	
Hepatocellular Carcinoma	miR-599/MYC	Suppressor	Tian et al. (2016)	
Hepatocellular Carcinoma	circ_0006916/miR-599/SRSF2	Suppressor	Zhu et al. (2020)	
Osteosarcoma	circ_0001721/miR-599	Suppressor	Li et al. (2019)	
Papillary Thyroid Carcinoma	miR-599/Hey2	Suppressor	Wang et al. (2020)	
0030788	607	Cervical Cancer	LncRNA-TP73-AS1/miR-607/CCND2	Suppressor	Zhang et al. (2019)	
Chronic Lymphocytic Leukemia	circ-CBFB/miR-607 /FZD3	Suppressor	Xia et al. (2018)	
Lung Squamous Carcinoma Cells	miR-607/CANT1	Suppressor	Qiao et al. (2021)	
Osteosarcoma	LINC00607/miR-607/E2F6	Suppressor	Zheng et al. (2020)	
Pancreatic Cancer	LINC01559/miR-607/YAP	Suppressor	Lou et al. (2020)	
Prostate Cancer	miR-607/BLM	Suppressor	Chen et al. (2019)	
0000227	647	Cervical Cancer	LncRNA-ZNFX1-AS1/miR-647	Suppressor	Yang et al. (2020)	
Colorectal Cancer	miR-647/NFIX	Promotor	Liu et al. (2017a)	
Gastric Cancer	miR-647/TP73	Promotor	Zhang et al. (2018a)	
Gastric Cancer	miR-647/ANK2, FAK, MMP2, MMP12, CD44, SNAIL1	Suppressor	Cao et al. (2017)	
Gastric Cancer	miR-647/SRF/MYH9	Suppressor	Ye et al. (2017)	
Gastric Cancer	LncRNA-PROX1-AS1	Suppressor	Song, Bi & Guo (2019)	
Gastric Cancer	miR-647/ANK2	Suppressor	Cao et al. (2018)	
Glioma	miR-647/HOXA9	Suppressor	Qin et al. (2020)	
Non-Small Cell Lung Cancer	miR-647/IGF2	Suppressor	Jiang, Zhao & Yang (2021)	
Non-Small Cell Lung Cancer	miR-647/TRAF2	Suppressor	Zhang et al. (2018b)	
Osteosarcoma	circ_0001649/miR-647	Promotor	Sun & Zhu (2020)	
Ovarian Cancer	circ-FAm53B/miR-647/MDM2	Suppressor	Sun, Liu & Zhou (2019)	
Prostate Cancer	NF-KappaB/circNOLC1/miR-647/PAQR4	Suppressor	Chen et al. (2020a)	
0000227, 0036592	767-3p	Glioma	miR-767-3p	Suppressor	Kreth et al. (2013)	
Hepatocellular Carcinoma	circ_0000673/miR-767-3p/SET	Suppressor	Jiang et al. (2018)	
Lung Adenocarcinoma	miR-767-3p/CLDN18	Suppressor	Wan et al. (2018)	
Non-Small Cell Lung Cancer	circ_0018818/miR-767-3p	Suppressor	Xu et al. (2020)	
0000229	935	Gastric Cancer	miR-935/SOX7	Promotor	Yang et al. (2016)	
Gastric Signet Ring Cell Carcinoma	miR-935/Notch1	Suppressor	Yan et al. (2016)	
Glioblastoma	miR-935/FZD6	Suppressor	Zhang et al. (2021)	
Glioma	miR-935/HIF1 α	Suppressor	Huang et al. (2020)	
Non-Small-Cell Lung Cancer	miR-935/SOX7	Suppressor	Peng et al. (2018)	
Non-Small-Cell Lung Cancer	miR-935/E2F7	Suppressor	Wang et al. (2019a)	
Osteosarcoma	miR-935/HMGB1	Suppressor	Liu et al. (2018)	
Liver Cancer	miR-935/SOX7	Promotor	Liu et al. (2017b)	
0030788	1248	Non-Small-Cell Lung Cancer	miR-1248/TYMS	Promotor	Xu et al. (2014)	

Figure 3 MiRNA pathway enrichment and target genes identification.

Significant signaling pathways of the 15 miRNAs utilizing the DIANA-miRPath (A), chromosome distribution of DEGs from GEPIA2 (B), and the Venn graph showing the 1076 overlapped target genes identified with intersection of three gene sets (C).

Obtaining target mRNAs in the ceRNA network

To forecast the target genes of these miRNAs, miRWalk 2.0, mirDIP v4.1, and GEPIA2 were utilized in this study. A total of 7688 mRNAs were validated or predicted by at least seven algorithms in miRWalk 2.0. A total of 3370 genes were forecasted by more than 10 algorithms in mirDIP v4.1 to be targets of those miRNAs. In GEPIA2, 7652 genes were identified as significant DEGs with P < 0.01 and —Log2(fold change)—> 1. The chromosomal distribution of DEGs is displayed in Fig. 3B. Thereafter, through the intersection of all three gene sets, we obtained a total of 1076 target mRNAs involved in the ceRNA network (Fig. 3C), whose overall expression levels were not significantly correlated with genders (Fig. S1).

Function enrichment analyses

To explore the potential functions of the 1076 target genes, GO and KEGG signal pathway enrichment analyses were performed using STRING. In terms of biological processes, these target genes covered majority of the hallmark pathways of tumors, including cell growth, cell cycle, proliferation, differentiation, migration, apoptosis, cell death, immune response, angiogenesis, as well as some well-known cancer-related pathways such as the Wnt and TGFβ signaling pathways, which were also exhibited in KEGG enrichment results. In addition, KEGG also showed enrichment in some other essential pathways, such as the cell cycle, p53, Hippo, MAPK, and stemness regulating pathways. The compositions of the target genes encompassed a wide range of cellular components and molecular functions from cytosol to synapse as well as from DNA binding to enzyme binding. Visualization of the enrichment results is displayed in Fig. 4. Taken together, functional enrichment results indicated that the ceRNA network is extensively involved in the pathogenesis of gliomas.

Figure 4 Functional enrichment analysis of target mRNAs in genes sets of biological process (A), cellular component (B), molecular function (C), and KEGG pathways (D).

The color intensity of the nodes shows the degree of enrichment of this analysis. Strength is the ratio between observed counts and the expected matching counts for a random list of the same size. The dot size represents the count of genes in a pathway.

Identification and assessment of hub genes

Utilizing gene expression data and corresponding clinical data from the TCGA-GBMLGG data set, we applied LASSO regression analysis to the 1076 target genes and obtained 46 significant candidate genes which were further used for multivariate Cox regression analysis (Fig. 5). Eventually, 11 independent prognosis-related hub genes were identified, including ARHGAP11A, DRP2, HNRNPA3, IGFBP5, IP6K2, KLF10, KPNA4, NRP2, PAIP1, RCN1, and SEMA5A. Thereafter, we checked the human protein atlas database to confirm the protein expression level changes of the 11 hub genes. Three of them (DRP2, IGFBP5, KLF10) are not provided with protein expression information. Two (ARHGAP11A, NRP2) of them showed no big difference between normal and glioma samples. The other six of them (HNRNPA3, IP6K2, KPNA4, PAIP1, RCN1, SEMA5A) showed significantly increased protein expression, consistent with our results (Fig. S2). The genes with supporting protein level results could be of more importance for further future validation study. But given the limited sample size of this part of data, we still took all those 11 genes as our subjects of analysis.

Figure 5 Forest plot of the multivariate Cox regression analysis result of LASSO significant genes utilizing gene expression data and corresponding clinical data from the TCGA-GBMLGG data set. Genes with P < 0.05 were considered significant.

Based on the expression value and correlation coefficients, we integrated these 11 genes as a whole signature and computed the total risk score of each sample to divide glioma patients into high- and low-risk groups. The K-M curve analysis indicated that the overall survival of the high-risk group was significantly shorter than that of the low-risk group (Fig. 6A). ROC curve analysis further showed decent sensitivity and specificity of this signature in predicting the prognosis of glioma patients (Fig. 6B). An external independent CCGA dataset was introduced to validate the results of the K-M curve and ROC curve analyses, and similar conclusions were obtained (Figs. 6C–6D).

Figure 6 Kaplan–Meier survival curves analysis and ROC curves analysis.

Kaplan–Meier survival curves of risk groups in the training dataset (A) and validation dataset (C), as well as the ROC curves of the hub gene signature in the training set (B) and validation set (D).

Construction of a circRNA–miRNA–mRNA network

To present the relationship between circRNA, miRNA, and mRNA, a ceRNA network consisting of seven circRNAs, 15 miRNAs, and 46 mRNAs was constructed using Cytoscape as shown in Fig. 7. The five upregulated circRNAs influenced nine miRNAs and 13 mRNAs, while the two downregulated circRNAs targeted seven miRNAs and eight mRNAs. Meanwhile, 25 mRNAs were under the two-way regulation of both groups of circRNAs.

Figure 7 A network of circRNA/miRNA/mRNA in glioma.

Oval, arrow, diamond, and octagon represents circRNA, miRNA, mRNA, and hub mRNAs respectively. For circRNAs, red indicates upregulation and blue means downregulation. Gradual color changes of mRNAs represent differences in the expression levels.

Immune cell infiltration features of prognosis-related genes

Gene expression profiles from TCGA-GBMLGG dataset and immune cell signatures from a previous study were used for immune infiltration analysis with the GSVA package (Bindea et al., 2013; Hanzelmann, Castelo & Guinney, 2013). Spearman correlation between infiltration levels of 24 immune cells and 46 prognosis-related genes demonstrated that their expression levels were closely related to tumor microenvironment immune infiltration (Fig. 8).

Figure 8 Correlation between 24 immune cells infiltration level and 46 LASSO regression analysis significant genes.

Candidate compounds from CMap and assessment

Notably, in order to explore the practical value of this study, the candidate compounds that might have effects on gliomas were predicted by CMap with the hub genes we screened out (Table 3). Based on the enrichment correlation coefficient, drugs such as fulvestrant, tanespimycin, mifepristone, tretinoin, and harman are the most promising potential therapeutic options for gliomas. With GSEA analysis on RNA-seq data from GEO (GSE59262 for mifepristone; GSE141789, GSE17227, and GSE61002 for tretinoin), mifepristone and tretinoin were proven to inhibit cell cycle and DNA repair pathways (Fig. 9). These novel therapeutic options would require more preclinical and clinical studies for further validation.

Table 3 Potential therapeutic options forecasted by CMap.

cmap name	dose	cell	score	up	down	instance_id	
fulvestrant	10 nM	PC3	−1	−0.212	0.661	4462	
tanespimycin	1 µM	PC3	−0.999	−0.407	0.465	1218	
mifepristone	9 µM	HL60	−0.983	−0.433	0.425	1569	
tretinoin	1 µM	PC3	−0.955	−0.276	0.558	1211	
harman	18 µM	PC3	−0.953	−0.495	0.337	4584	
miconazole	10 µM	HL60	−0.941	−0.453	0.368	1977	
ifosfamide	15 µM	PC3	−0.933	−0.26	0.554	5805	
trimethylcolchicinic acid	12 µM	PC3	−0.931	−0.457	0.356	4202	
rifampicin	5 µM	PC3	−0.909	−0.281	0.512	4008	

Figure 9 The impact of candidate drugs on glioma cells.

mRNA expression profiles of glioma cells treated with drug(mifepristone/tretinoin) or vehicle were analyzed with GSEA. NES, normalized enrichment score; FDR, false discovery rate. Negative value of NES means inhibition; positive value means promotion. FDR <0.25 were considered as significant.

Discussion

CircRNA can act as a molecular sponge for miRNAs to de-repress all target genes of these miRNAs (Kristensen et al., 2019; Meng et al., 2017). Accumulating evidence has shown that this circRNA-miRNA-mRNA network plays an important role in the pathogenesis of gliomas, encompassing a wide range of phenotypes, such as proliferation, migration, and invasion (Chen et al., 2019; Ding et al., 2020; Wang et al., 2018). Therefore, circRNAs and miRNAs are increasingly regarded as promising therapeutic targets or diagnostic biomarkers. GBM can be divided into different subsets with diverse responses to various therapies as a group of heterogeneous intracranial neoplasms with distinct histopathological and molecular biological characteristics (Lapointe, Perry & Butowski, 2018). Thus, there is an urgent need to establish reliable risk stratification methods to classify GBM patients into various risk groups to benefit from various treatment strategies. Many studies have explored the prognostic signatures of gliomas in the context of epigenetic modifications or lncRNAs (Niu et al., 2020; Pan et al., 2021). However, so far, there has been no comprehensive and in-depth study of the molecular signatures of circRNA-related ceRNA networks in gliomas. Therefore, in this study, we constructed a circRNA–miRNA–mRNA network in glioma to help understand its pathogenesis as well as aid in risk stratification and therapeutic decision-making.

In this study, eight DECs (hsa_circ_0001156, hsa_circ_0030788, hsa_circ_0034182, hsa_circ_0000227, hsa_circ_0018086, hsa_circ_0000229, hsa_circ_0036592, and hsa_circ_0002765) were identified as DECs in the first step. To the best of our knowledge, all of them were found to be abnormally expressed in glioma for the first time and have not been studied so far, which makes them potential novel biomarkers or therapeutic targets.

Seven out of eight circRNAs (except for hsa_circ_0001156) were identified as ceRNAs to bind 15 miRNAs (hsa-miR-1200, hsa-miR-1205, hsa-miR-1248, hsa-miR-1303, hsa-miR-3925-5p, hsa-miR-5693, hsa-miR-581, hsa-miR-586, hsa-miR-599, hsa-miR-607, hsa-miR-640, hsa-miR-647, hsa-miR-6867-5p, hsa-miR-767-3p, and hsa-miR-935). As for hsa_circ_0001156, it might still be involved in the pathogenesis of gliomas through functions other than miRNA sponges, such as coding proteins, interacting with RNA binding proteins, or modulating the stability of mRNAs.

As shown in Table 2, among the 15 miRNAs identified, miR-581, miR-586, and miR-1248 were reported to promote tumor progression, while miR-599, miR-607, and miR-767-3p were shown to be tumor suppressors in various cancers. Some studies have shown contradictory results regarding the effects of miR-1303, miR-647, and miR-935 on some tumors. This could partly result from differences in the cell lines used, the phenotypes selected, or the pathways studied. However, further research is needed to resolve these discrepancies. Nevertheless, it is noteworthy that miR-647, miR-767-3p, and miR-935 reportedly suppress the progression of glioma through multiple regulatory axes, including miR-647/HOXA9, miR-935/FZD6, miR-935/HIF1 α, or miR-767-3p itself, making them promising biomarkers and therapeutic targets for glioma. In addition, the other six miRNAs without previous studies on tumors (miR-1200, miR-1205, miR-3925-5p, miR-5693, miR-640, and miR-6867-5p) could also be novel fields worth exploring, which could possibly lead to unexpected discoveries.

Given the correlation between miRNAs expression levels and their biological functions, we further checked their expression levels in glioma tissues with CGGA data and GEO datasets, as shown in (Tables S2–S3). According to those two datasets, the miR607 and miR587 have relatively lower expression levels, implying they could be less important in glioma studies. And miR581 has relatively higher expression, giving it higher importance for glioma. But it is noteworthy that different datasets showed considerably different expression of those miRNAs, probably due to the different platforms used and various samples chosen. And their expression values vary quite a lot from sample to sample even for the same dataset. So, their actual expression levels still await to be examined.

CircRNAs fulfill their functions by de-repressing the target genes of miRNAs. Therefore, to further explore the effects of circRNAs on glioma, 1076 overlapping target genes were collected and used for functional enrichment analyses. The ten major hallmarks of tumors include sustaining proliferative signaling, evading growth suppressors, resisting cell death, enabling replicative immortality, inducing angiogenesis, activating invasion and metastasis, reprogramming of energy metabolism, evading immune destruction, genome instability and mutation, and dysregulating cellular energetics (Hanahan & Weinberg, 2011), most of which were covered in our enrichment result of biological processes. This indicates that the ceRNA network we built here is extensively involved in the initiation and progression of gliomas.

The pathway enrichment results uncovered the involvement of many essential tumor-related pathways such as the Wnt, TGFβ, cell cycle, p53, Hippo, MAPK, and stemness regulating pathways. Wnt signaling, commonly divided into β-catenin-dependent (canonical) and independent (non-canonical) signaling, is one of the key cascades regulating development (Klaus & Birchmeier, 2008). Its role in carcinogenesis has mostly been described in colorectal cancer, along with some other cancer entities (Zhan, Rindtorff & Boutros, 2017). The transforming growth factor (TGF)-β signaling pathway is deregulated in many diseases and has dual functions in cancers. It suppresses tumors in healthy cells and early stage cancer cells but promotes tumorigenesis, metastasis, and chemoresistance in late-stage cancer (Colak & Ten Dijke, 2017). p53 is a tumor suppressor protein that regulates cell growth by promoting apoptosis and DNA repair under stressful conditions (Kanapathipillai, 2018). The Hippo pathway largely consists of a kinase cascade (MST1/2 and LATS1/2) and downstream transcriptional coactivators (YAP and TAZ), controlling transcriptional programs involved in cell proliferation, survival, mobility, stemness, and differentiation (Ma et al., 2019). The MAPK/ERK pathway is a chain of proteins that communicate signals from a receptor on the cell surface to the DNA in the nucleus of the cell (Orton et al., 2005). Alteration of this pathway is often a necessary step in the development of many cancers (Drosten & Barbacid, 2020). Cancer stem cells are capable of sustaining tumors by aiding metastasis, therapy resistance, and tumor microenvironment maintenance, making the stemness regulation key traits and mechanisms for tumor progression (Saygin et al., 2019). All of these pathways, which were under the control of the ceRNA network we constructed in this study, have been shown to participate in the initiation or progression of gliomas (He et al., 2019; Lan et al., 2019; Lee et al., 2017; Ma et al., 2018; Masliantsev, Karayan-Tapon & Guichet, 2021; Zhao et al., 2019). Altogether, these 1076 target genes, regulated indirectly by the circRNAs identified in the present study, play essential roles in the pathogenesis of gliomas.

Thereafter, LASSO regression analysis and multivariate Cox regression analysis were applied to the 1076 genes consecutively. Forty-six LASSO significant genes and 11 independent prognosis-related hub genes were identified (ARHGAP11A, DRP2, HNRNPA3, IGFBP5, IP6K2, KLF10, KPNA4, NRP2, PAIP1, RCN1, and SEMA5A). Among them, ARHGAP11A, DRP2, HNRNPA3, KLF10, PAIP1, and RCN1 have not yet been studied in gliomas. Three mRNAs were identified as oncogenes in gliomas, which was consistent with our multivariate Cox regression analysis result: IGFBP5 can increase cell invasion and inhibit cell proliferation via the EMT and Akt signaling pathways in GBM (Dong et al., 2020); IP6K2 was reported to promote cell proliferation and inhibit cell apoptosis under the regulation of the LINC00467/miR-339-3p axis (Liang & Tang, 2020); and KPNA4 is capable of facilitating epithelial-mesenchymal transition in glioma, which can be suppressed by miR-181b, a tumor-suppressive miRNA (Wang et al., 2015). Surprisingly, the roles of the other two mRNAs in glioma were shown to be different from our analysis result: NRP2 promoted glioma cell growth, invasion, and angiogenesis (Zheng et al., 2013); and SEMA5A, whose expression is markedly reduced in higher grades of glioma, can impede motility and promote differentiation of human gliomas (Li & Lee, 2010). This discrepancy might result from the fact that we used survival data for analysis, and those genes were studied only in vitro for some specific phenotypes, while the in vivo result of survival might be influenced by multiple other conditions and phenotypes (such as immune response and therapy sensitivity). However, the functions of these genes require further experimental verification. Nevertheless, the K-M curve analysis and ROC curve analysis in both training and external validation datasets proved that these genes together are a competent signature for predicting the prognosis of gliomas.

In addition, we checked the human protein atlas database to confirm the protein expression level changes of the 11 hub genes. Three of them (DRP2, IGFBP5, KLF10) are not provided with protein expression information. Two (ARHGAP11A, NRP2) of them showed no big difference between normal and glioma samples. The other six of them (HNRNPA3, IP6K2, KPNA4, PAIP1, RCN1, SEMA5A) showed significantly increased protein expression, consistent with our results (Fig. S2). This does not mean that DRP2, IGFBP5, KLF10, ARHGAP11A and NRP2 are not essential genes for glioma pathogenesis, given the missing data and limited sample size we found. But still this could give us some hints that those genes with supporting HPA results (namely HNRNPA3, IP6K2, KPNA4, PAIP1, RCN1, SEMA5A) could be more promising targets for future further validation studies.

Recently, circRNAs were reported to play a significant role in multiple immune-related biological processes, including innate and adaptive immune responses, immune cell homeostasis, immune recognition, and anti-tumor immunity (Yan & Chen, 2020; Zhang et al., 2020). Comprehensive recognition of circRNA-mediated immune cell infiltration in glioma can provide novel insights into risk stratification and clinical therapeutic strategies. Hence, we profiled tumor microenvironment immune cell infiltration utilizing 46 prognosis-related genes from the LASSO regression analysis as shown in Fig. 8. The current consensus is that the anti-tumor immune response in glioma is largely suppressed by brain-resident microglial cells and bone marrow-derived macrophages, and is mainly promoted by CD8+ T cells (Pinton et al., 2019). However, the roles of other immune cells, such as B cells, are still debatable (Pinton et al., 2019; VonRoemeling et al., 2020). Our results showed that generally the genes that were positively related to macrophages were negatively correlated with CD8+ T cells, and vice versa. This indicated that these genes could play considerable roles in immune infiltration switch of the tumor microenvironment. However, it should be noted that there is still a lack of systemic immune cell markers for gliomas. Existing immune markers are mostly constructed in other tumors, and some glioma-specific immune cells (such as microglial cells) still lack convincing specific markers. Specific immune cell markers for gliomas are urgently needed for a robust assessment of immune infiltration and understanding of immune response mechanisms in gliomas.

Glioma is one of the most drug-resistant malignancies with frequent recurrence after chemotherapy, making it necessary to explore novel compounds or drugs that may have a therapeutic effect. Here, with the hub genes identified, some potential drugs were acquired from the CMap database. Although the prediction of CMap was mostly based on experiments in prostate cancer and leukemia cell lines, the effects of these drugs have also been verified in multiple other tumors. For instance, fulvestrant is a selective estrogen receptor degrader that has been extensively studied for its therapeutic effects in breast cancer (Slamon et al., 2020). Harmane is a tremorigenic β-carboline capable of inhibiting mitochondrial viability and increasing reactive oxygen species levels (Khan, Patel & Kamal, 2017). Its semi-synthetic derivative, B-9-3, showed an anti-proliferative effect in lung cancer, breast cancer, and colorectal carcinoma cell lines via induction of apoptosis and inhibition of cell migration (Daoud et al., 2014). Some of these drugs have been proven to interfere with the progression of gliomas. Tanespimycin is a well-characterized HSP90 inhibitor that can inhibit the growth of GBM and synergize with radiation (Sauvageot et al., 2009). Mifepristone was reported to be a potential therapy for reducing angiogenesis and TMZ resistance in GBM (Llaguno-Munive et al., 2020). Tretinoin, an all-trans retinoic acid, was shown to significantly induce apoptosis and suppress stemness in GBM (Chen et al., 2014; Hu et al., 2017). Importantly, mifepristone and tretinoin were shown to inhibit cell cycle and DNA repair of glioma according to our own GSEA analysis result (Fig. 9). Given that radiation and temozolomide, the major non-surgical treatments for glioma, both work through inducing DNA damage, those novel drugs could be promising supplementary therapeutic treatments which can be applied in combination with radiotherapy or chemotherapy.

Several limitations of this study should be considered. The construction of circRNA/miRNA/mRNA regulatory networks and the prediction of therapeutic drugs largely relied on a series of bioinformatics algorithms and databases, whose authenticity and accuracy still await the verification of numerous experiments. Therefore, we adopted and integrated multiple databases for all predictions in the present study to improve robustness. In addition, the retrospective research design could display some statistical bias and the traditional bulk sequence transcriptome data would lack comprehensive exploration of intra-tumoral heterogeneity. A prospective study design and utilization of single-cell omics techniques will help address this issue and provide more accurate and reliable results in the future. However, based on circRNA/miRNA/mRNA regulatory networks, we established a superior predictive signature to assess the clinical outcomes of patients with GBM and forecasted some promising candidate drugs.

Conclusions

Through the construction of a circRNA/miRNA/mRNA regulatory network in glioma and the combination of survival analysis, this study successfully identified 11 circRNA-related mRNA signatures to predict the prognosis of GBM patients. Additionally, we determined that circRNA-regulated hub genes were correlated with specific immune cell infiltration levels and proposed some potential therapeutic options. Comprehensively exploring the circRNA/miRNA/mRNA regulatory network in GBM will enhance our understanding of the pathogenesis and immune infiltration features of glioma, promote treatment strategies, and improve clinical outcomes.

Supplemental Information

Supplemental Information 1 Codes used to generate figures

Click here for additional data file.

Supplemental Information 2 CirRNA expression level in the datasets studied in this paper

Click here for additional data file.

Supplemental Information 3 Target miRNA expression level in CCGA 198 samples miRNA seq dataset

Click here for additional data file.

Supplemental Information 4 GSE165286-NT vs GBM targets micro RNA expression level-FPKM

Click here for additional data file.

Supplemental Information 5 CGGA sample ID list

Click here for additional data file.

Supplemental Information 6 TCGA sample ID list

Click here for additional data file.

Supplemental Information 7 No gender correlation of target mRNAs

(A) PCA plot of gene expression level between males and females. (B) volcano plot of differentially expressed genes between males and females.

Click here for additional data file.

Supplemental Information 8 The Human Protein Atlas images of the 6 hub genes with consistent supporting results

Click here for additional data file.

Additional Information and Declarations

Competing Interests

Author Contributions

Data Availability

The authors declare there are no competing interests.

Yi He conceived and designed the study, acquired and analyzed the data, prepared figures and tables, authored and reviewed drafts of the paper, and approved the final draft.

Yihong Chen acquired and analyzed the data, prepared figures and tables, and approved the final draft.

Yuxin Tong analyzed the data, prepared figures and tables, and approved the final draft.

Wenyong Long and Qing Liu conceived and designed the experiments, authored and reviewed drafts of the paper, and approved the final draft.

The following information was supplied regarding data availability:

The codes used for bioinformatics analysis are available in the Supplementary File.

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
