# Peer review of "Identification of a circRNA-miRNA-mRNA regulatory network for exploring novel therapeutic options for glioma"

_PeerJ, doi:10.7717/peerj.11894_

## Round 0.1 · original submission · Minor Revisions

All three reviewers have positive suggestions for your current manuscripts. Still, there are some questions that need to be addressed as raised by the reviewers.

Reviewer 1 ·

Basic reporting

See General Comments

Experimental design

See General Comments

Validity of the findings

See General Comments

Additional comments

In this paper, He and colleagues constructed a circRNA–miRNA–mRNA network in glioma with GEO, GEPIA2, circBank, CSCD, CircInteractome, mirWalk 2.0, and mirDIP 4.1. The present study emphasizes the potential pathogenesis of the circRNA-miRNA-mRNA which may therapeutic options for glioma. In general, this manuscript was well written and well designed. Nevertheless, I have the following comments for the authors to address before accepting for publication.
1. Dose sex difference was observed in the circRNA-miRNA-mRNA regulatory network on glioma?

·

Basic reporting

HE and Chen et. al used multiple analysis methods constructed an interesting network in glioma, which provides potential therapeutic options for tackling glioma. The analyzes are rational and comprehensive and the results also make sense such as functional enrichment analyses revealed a lot of hallmarks of tumorigenesis. This study provides a good benefit to the glioma field. I only have the following minor issues:

1. The biggest concern is that, also pointed out by the authors, there is no experimental verification. It is understandable that it is difficult to investigate glioma in human or even beyond the scope of this paper, but it would be really great to verify either the effect of the drugs using cultured cells or rodent models. Or it will be really interesting to take a look at the amount of the relevant circRNA and ceRNA in glioma tissue. Or is there any relevant reported data to verify the results?

2. The paper is generally well written. The intro and background are clear. The results are written clearly but fluency should be improved a bit. Also some expressions need to be improved such as line 56 (‘that hope to’), grammar error such as line 67 (‘the’), and also please minimize the usage of line 297 ‘As we all know’.

Experimental design

no comment

Validity of the findings

no comment

·

Basic reporting

The study is well conducted and the manuscript is written in an unambiguously and professional way. However, there are a few questions/problems the authors need to solve:

1. The current study investigated 8 circRNAs but why only 6 of them are present in figure 2c?

2. The expression of RCN1 gene is not labeled as significant difference in figure 5.

3. Please give a brief introduction to explain the importance of assessing immune cell infiltration for the prognosis of glioma.

4. more references should be added in the introduction, results, and discussion for some sentences:
Introduction:
(1) “Gliomas comprise majority of primary intracranial neoplasms with high heterogeneity…”
(2) “Recent advances in precision medicine, genomics, immunology, and other disciplines have uncovered…”
(3) “Circular RNA (circRNA) is a type of non-coding RNA derived from the exon or intron…”
Results:
“Gene expression profiles from TCGA-GBMLGG dataset and immune cell signatures from a previous study …”
Discussion:
(1) “CircRNA can act as a molecular sponge for miRNAs to de-repress all target genes …”
(2) “Wnt signaling, commonly divided into β-catenin-dependent (canonical) and …”
(3) “The MAPK/ERK pathway is a chain of proteins that communicate …”
(4) “The current consensus is that the anti-tumor immune response in glioma is …”
(5) “Harmane is a tremorigenic β-carboline capable of inhibiting mitochondrial viability …”
(6) “Importantly, mifepristone and tretinoin were shown to inhibit cell cycle and DNA repair …”

Experimental design

Generally, the experiments are well designed but more details should be added to the methods in order to enhance the reproducibility:

Please provide a detailed description of how to integrate the 11 hub genes as a whole signature and compute the total risk score of each sample. Additionally, what are the criteria that separate the glioma patients into high and low-risk groups based on the risk score?

Validity of the findings

1. The expression levels of the circRNAs and miRNAs are critical for their biological functions. Please provide the expression levels of the circRNAs and miRNAs in the tissues that are investigated in this study.

2. Please provide the exact ID numbers of the TCGA-GBMLGG dataset that is used for the assessment of immune cell infiltration.

3. Please provide the exact ID numbers of the CCGA dataset that is used for the assessment of hub genes.

4. The description in the discussion is not consistent with the finding in the current study:
“Our results showed that the selected genes were closely related to immune infiltration and had generally opposite correlations with macrophages …”

Additional comments

1. Typo in the abstract (page 5 line 34): ceNRA > ceRNA

---

## Round 0.2 · accepted · Accept

Congratulations! As all concerns from the reviewers were substantially addressed in the revised manuscript. All reviewers agreed to publish your beautiful work in PeerJ.

Reviewer 1 ·

Basic reporting

.

Experimental design

.

Validity of the findings

.

Additional comments

I recommend this manuscript to be accepted for publication as all my concerns has been addressed.

·

Basic reporting

He and colleagues substantially revised their manuscript according to reviewers' comments. I think it is now suitable for publishing in PeeJ now.

Experimental design

None

Validity of the findings

None

·

Basic reporting

The authors have satisfactorily resolved all the questions I asked, and I think the manuscript is good to be published.

Experimental design

No comment

Validity of the findings

No comment